# Sleep and cognition in South African patients with non-functioning pituitary adenomas

Olivia de Villiers[1], Claudia Elliot-Wilson[1], Kevin G. F. Thomas[1], Patrick L. Semple[2‡], Thurandrie Naiker[3‡], Michelle Henry[4*], Ian L. Ross[5]

1 ACSENT Laboratory, Department of Psychology, University of Cape Town, Cape Town, South Africa, 2 Division of Neurosurgery, University of Cape Town, Cape Town, South Africa, 3 Department of Radiation Oncology, University of Cape Town and Groote Schuur Hospital, Cape Town, South Africa, 4 Numeracy Centre, University of Cape Town, Cape Town, South Africa, 5 Division of Endocrinology, University of Cape Town, Cape Town, South Africa

☉ These authors contributed equally to this work.
‡ PLS and TN also contributed equally to this work.
* m.henry@uct.ac.za

**Data Availability Statement:** The data are available on ZivaHUb at https://figshare.com/s/f31b8954618a5e801532

## Abstract

Strong lines of evidence in the neuroscience literature indicate that (a) healthy sleep facilitates cognitive processing, and (b) sleep disruption is associated with cognitive dysfunction. Despite the fact that patients with pituitary disease often display both disrupted sleep and cognitive dysfunction, few previous studies investigate whether these clinical characteristics in these patients might be related. Hence, we explored whether sleep disruption in patients with pituitary disease mediates their cognitive dysfunction. We recruited 18 patients with non-functioning pituitary adenomas (NFPA) and 19 sociodemographically matched healthy controls. They completed the Global Sleep Assessment Questionnaire (thus providing self-report data regarding sleep disruption) and were administered the Brief Test of Adult Cognition by Telephone, which assesses cognitive functioning in the domains of processing speed, working memory, episodic memory, inhibition, and reasoning. We found no significant differences in cognition between patients and controls. Furthermore, spectra of sleep disturbance did not differ significantly between patients and controls. Our data suggest that NFPA patients' cognition and sleep quality is relatively intact, and that sleep disruption does not mediate cognitive dysfunction. Larger studies should characterize sleep and cognition in patients with NFPA (and other pituitary diseases) to confirm whether disruption of the former mediates impairment in the latter.

## Introduction

Sleep is a universal behavioural and biological phenomenon that supports crucial aspects of human health [1]. Healthy sleep is associated with lower risk for certain medical illnesses (e.g., diabetes, coronary heart disease) and psychiatric disorders (e.g., depression), and improves concentration and overall cognitive performance [2, 3]. Hence, disrupted sleep can have detrimental consequences for physical, mental, and cognitive health. Patients with pituitary disease

**Funding:** The author(s) received no specific funding for this work.

**Competing interests:** The authors have declared that no competing interests exist.

provide an ideal opportunity to investigate relationships between sleep and cognition because the pituitary gland is a central component of the neuroendocrine system that regulates the sleep-wake cycle, and also plays a critical role in learning and memory processing [4]. The purpose of the study described here was to determine whether sleep disruption mediates cognitive dysfunction in patients with pituitary disease, or whether the appearance of sleep disturbances and cognitive impairments in those patients are unrelated (e.g., that they might emerge as independent disease-related characteristics).

Human sleep comprises two main stages: rapid eye movement (REM) and non-REM (NREM) sleep. The latter is further subdivided into stage 1 (light sleep), stage 2, and stage 3, which is characterized by deep sleep and slow wave sleep (SWS) on electroencephalogram, electroencephalogram and electromyography readings. Of particular interest here, because our population of interest are patients with pituitary disease, are bidirectional relations between hypothalamic-pituitary adrenal (HPA)-axis activity and sleep architecture. For instance, HPA-axis activity and consequent release of adrenocorticotropic hormone (ACTH) and cortisol are inhibited during SWS [5]. However, when the HPA axis is highly active (as happens, for example, when the organism is exposed to acute or chronic stress) and levels of circulating cortisol are thereby elevated, SWS is inhibited and sleep is fragmented. Such disrupted sleep patterns can exacerbate HPA-axis hyperactivity, resulting in a problematic feedback loop [6]. REM sleep is also affected by HPA-axis activity. Some accounts suggest that corticotropin releasing hormone, which promotes wakefulness, may inhibit REM sleep [7], and that cortisol may be a necessary element for REM sleep initiation [8, 9].

Although numerous cognitive processes are supported by healthy sleep, new learning, and memory consolidation and consolidation has been the primary focus of a large body of neuroscientific research [10, 11]. Research has shown that healthy sleep supports memory functioning in two ways: a good night's sleep helps prepare the brain for new learning the following day, and sleep-dependent memory consolidation helps strengthen new memories learned during the day [12]. Sleep plays an important role in restoring neuronal functioning for new learning on the subsequent day [13]. Several studies have confirmed this and shown that when sleep is deprived, memory encoding is disrupted [14]. Studies also suggest that an uninterrupted night of sleep prior to learning allows for more efficient retrieval of that information on subsequent occasions [15, 16]. Sleep-dependent memory consolidation on the other hand involves the movement of information encoded during the waking day into permanent memory traces. A key mechanism underlying this process occurs during SWS, when newly encoded information is transferred between hippocampal and neocortical regions in a reiterative process, thus making uninterrupted sleep after learning an important part of memory consolidation [17, 18]. Overall, this body of research suggests that sleep disruption occurring before or after daytime encoding of information will result in poorer memory for that information.

Although sleep-dependent memory consolidation is clearly a critical cognitive process, sleep is not the sole state during which recently acquired memory traces are transformed into stable representations and transferred to non-hippocampal brain regions. Daytime consolidation is possible, and is indeed probable during periods of quiet waking rest [19]. The mechanisms facilitating memory consolidation during sleep and waking are likely quite similar [20], and hence it is possible that when sleep-dependent memory consolidation is disrupted similar negative effects are imposed on daytime consolidation. These negative effects may be amplified when daytime waking rest states are interrupted by excessive attentional demands, poor attentional functioning, or lack of ability to be immersed in such states.

Pituitary disease refers to a group of disorders in which the pituitary gland functions abnormally, usually because of organic damage (e.g., by a tumour, radiation, or neoplasm), leading to hyper- or hypo-secretion of pituitary hormones. Pituitary tumours can be classified by size

(i.e., microadenomas [diameter <1 cm] or macroadenomas [diameter >1 cm] [21] or by functionality (i.e., functioning or non-functioning adenomas). Functioning adenomas directly affect hormone release from the pituitary gland, and consequently adversely influence normal physiology and cognition. Although non-functioning pituitary adenomas (NFPA) can be benign, they may also have adverse effects on physiological states (such as sleep) and on cognitive performance [22, 23].

Patients with NFPA often present with profound disturbances in normal sleep architecture (e.g., decreased REM sleep, increased stage 1 sleep, more night-time awakenings) and with increased daytime sleepiness [24–26]. These patients also often present with cognitive dysfunction, although the specific nature of the impairment varies depending on the nature of the organic damage [27, 28]. For instance, Yedinak and Fleseriu [29] found that patients with acromegaly self-reported poor learning and concentration abilities and showed difficulty maintaining focus on imminent tasks, whereas those with NFPA self-reported poor mental agility and impaired verbal memory.

Although cognitive deficits in patients with NFPA are most often attributed to the underlying pathology, treatment for the tumour may also be a contributing factor. The preferred treatment for patients with symptomatic NFPAs is transsphenoidal surgical resection. Radiotherapy is only recommended as primary treatment option when surgery is contraindicated; usually, it is used a postoperative adjuvant treatment in cases of regrowth or where the surgical result is suboptimal [21]. Noad et al. [30], comparing patients who had received both surgery and radiotherapy with those who had received only surgery for the removal of the pituitary tumours, reported that both groups produced impaired performance on a standard assessment of executive functioning. Brummelman et al. [27], comparing NFPA patients who had received both surgery and radiotherapy with those who had received only surgery, also found that both groups performed more poorly than normative standards on tests of learning, memory, and design fluency.

Only one published study has investigated possible associations between sleep disruption and cognitive dysfunction in pituitary disease. Wennberg et al. [31] administered self-report questionnaires (the Epworth Sleepiness Scale [32] and the Pittsburgh Sleep Quality Index [33] assessing sleep quality, and a comprehensive neuropsychological test battery assessing attention, working memory, visuospatial abilities, language, memory, and executive functioning via standard pen-and-paper tests, to 67 acromegalic patients. They found a significant association between sleep quality and cognitive performance (poorer sleep predicted worse cognitive performance) and concluded that the link between the two in patients with pituitary disease warrants further investigation. Limitations of this study were that it (a) sampled patients from only one type of pituitary disease, (b) did not include comparison with a matched healthy control group, and (c) could not, by design, undertake direct investigation of a possible mediational relationship.

## The current study

As explained in the review above, the pituitary gland is a central component of the neuroendocrine system that, via thyroid and glucocorticoid hormones, regulates the sleep-wake cycle and plays a critical role in memory processing. Hence, we wished to determine whether patients with non-functioning pituitary adenomas [NFPA] would present with compromised sleep quality and impaired cognitive performance, even when adequately replaced with hormone therapy. We also wished to determine, given known relationships (described above) between sleep and cognition, whether disrupted sleep in those patients might partially account for their compromised cognitive functioning. To achieve those aims, we collected self-report data

regarding sleep quality and took objective measures of cognitive performance from a group of NFPA patients and from a group of sociodemographically matched healthy controls. This case-controlled design allowed us to test these specific hypotheses: (1) patient performance on an objective test of cognitive performance will be significantly worse than that of controls; (2) patients will self-report significantly more disrupted sleep than controls; and (3) sleep disturbance will mediate the relationship between the presence of pituitary disease and cognitive performance.

## Materials and methods

### Participants

We recruited 18 patients with NFPA (9 women; see Fig 1). The NFPA must have been confirmed by medical records, and patients were required to have been on a stable treatment replacement regimen for at least 3 months prior to study enrolment. They were also required to have received either radiotherapy or surgery, or a combination, to be considered for enrolment.

Patients with NFPA were excluded from participation if they had a medical comorbidity (e.g., HIV, epilepsy) [34, 35] that is known to have independent effects of cognitive functioning.

We also recruited 19 healthy controls (8 women). They were spouses or family members of patients and other community-dwelling individuals. Because there are well documented data describing (a) age-related variability in cognition and in sleep patterns [36–39], (b) sex differences in sleep patterns and sleep disorders [40], (c) protective effects of education against cognitive decline and dysfunction [41], and (d) positive associations between socioeconomic status (SES) and cognitive performance [42], we attempted to match each control to a patient on age (within 5 years), sex, level of education (within 2 years), and SES.

Our general eligibility criteria, for both patients and controls, included age between 18 and 65 years and absence of neurological, psychiatric, medical, or other illness/disease which may negatively affect cognitive functioning or alter sleep quality.

### Materials and procedure

We telephoned potential participants and invited them to enroll in the study. If they accepted in principle, we gave them full details of the protocol and of the requirements associated with their participation and asked for their verbal consent. If they provided consent, we asked if we could begin administration of the study materials during the same telephone call. If they were willing to participate but could not complete the questionnaires during the initial call, we scheduled an alternative time.

The first instrument we administered was a study-specific questionnaire that collected data on age, sex, education, SES, and medical history (including, where relevant, diagnosis, duration of illness, and treatment type and duration). Next, we administered the *Global Sleep Assessment Questionnaire (GSAQ)* [43], a reliable and well-validated 11-item instrument that measures sleep disruption. Each item asks about a possible symptom the respondent may have experienced in the previous 4 weeks and is answered using a 4-point scale (with response options of *always*, *usually*, *sometimes*, and *never*). Hence, the possible score range is 11–44, with lower values indicating greater possibility of the presence of a sleep disorder. The instrument's developers report that it has good internal consistency ($\alpha = 0.76$), test-retest reliability ranging from 0.51 to 0.92 over a period of 7–14 days, and good convergent and divergent validity. Although the GSAQ is normally a self-administered in-person questionnaire, its length and layout lend themselves for adaptation to online or telephonic administration [44].

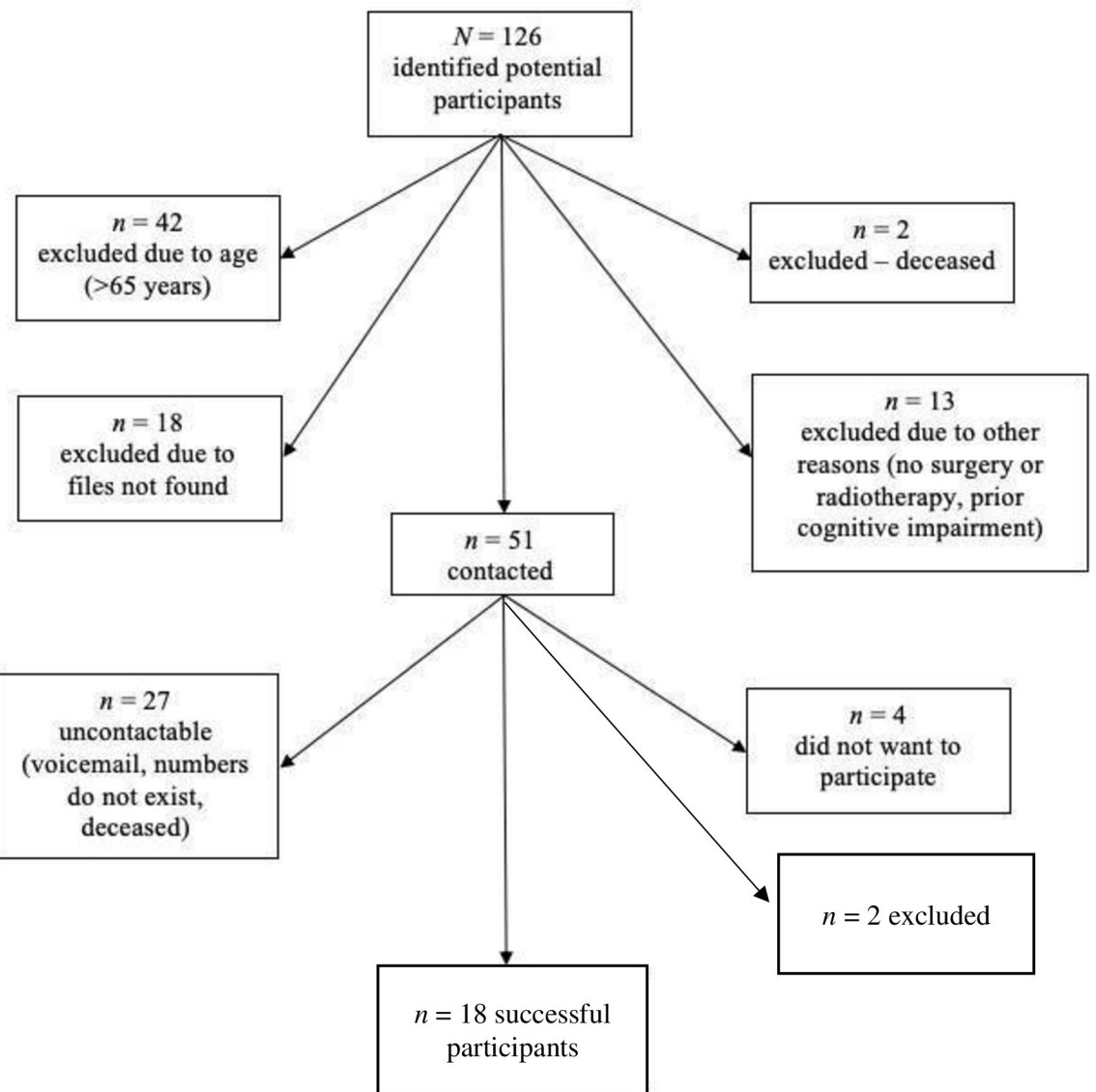

**Fig 1. Flowchart showing attrition during NFPA patient recruitment.** *Note.* We identified the initial pool of potential patient participants (*N* = 126) by inspecting the Groote Schuur Hospital Radiotherapy Clinic patient database.

Finally, we administered the *Brief Test of Adult Cognition by Telephone* (*BTACT)* [45];, a reliable and well-validated instrument that measures performance across several different cognitive domains. The BTACT has been used in research studies assessing both cognitively impaired and cognitively intact adults, across a wide range of ages and educational backgrounds [46, 47]. It comprises six subtests, all based on standard in-person psychometric tests, which have been adapted for telephonic administration.

Within the BTACT, the *Rey Auditory-Verbal Learning Test* assesses verbal memory for a 15-word list across immediate and delayed recall trials. The outcome measure for each trial is the number of words recalled correctly. The *Digits Backwards* subtest assesses working memory by requiring the participant to repeat sequences of numbers (starting with a 2-digit sequence and progressing to a maximum of an 8-digit sequence) in reverse order to that read

by the researcher. The outcome measure is how many digit sequences are repeated correctly. The *Category Fluency* subtest assesses verbal generativity by requiring the participant to name as many different animals as they can within 60 seconds. The outcome measure is how many different animals are named. The *Stop and Go Task* assesses the ability to inhibit automatic responses. One round of trials requires the participant to say 'go' when the researcher says 'green' and 'stop' when the researcher says 'red'. The second round requires the participant to say 'stop' when the researcher says 'green' and 'go' when the researcher says 'red.' The third round mixes these instructions, requiring the participant to alternate between the first- and second-round responses to green and red as the researcher indicates. The outcome measure is the number of correct responses in the third round. The *Number Series* subtest measures reasoning by asking the participant to complete the sequence of numbers the researcher has read out. The outcome measure is the number of correct answers across five trials. The *Backward Counting Task* measures processing speed by asking the participant to count backwards in 1's from 100. The outcome measure is how many numbers counted backwards correctly minus the number of errors made within 45 seconds.

Complete administration took approximately 45 minutes. At the end of the session, we invited participants to ask any questions and debriefed them following a standard verbal script. At the conclusion of all data collection procedures, we emailed each participant a debriefing letter.

## Data management and statistical analysis

**Deriving outcome variables.** We scored the GSAQ following standard methods [43].
Conventional BTACT scoring involves calculating a *z*-score for each subtest, then calculating an average of those to obtain a composite *z*-score. This composite serves as an index of global intellectual functioning. This method is potentially problematic in that it makes assumptions, which are often violated, about how individual subtest performance is related to overall cognition [45, 46]. Hence, Gavett et al. [48] proposed an alternative scoring method. Their bi-factor model generates both a general factor (i.e., one representing overall cognitive ability as measured by all the test items) and a secondary factor (i.e., one representing specific cognitive factors assessed by the different subtests). The bi-factor model produces an unadjusted overall *z*-score, in addition to *z*-scores adjusted via regression modeling for participants' age, sex, education, and combinations of the three. Gavett and colleagues present strong psychometric evidence suggesting that using this bi-factor method may be highly advantageous in interpreting BTACT performance. However, the conventional scoring method is still useful in that it permits comparisons among individual subtests. Hence, we used both scoring methods because the conventional scoring method allows profiling of cognitive performance across different domains, while the bi-factor method allows an understanding of global cognitive status.

**Inferential analyses.** We used R-Studio (v. 1.1.456) for all statistical analyses, with the threshold for statistical significance (α) set at 0.05. The primary analyses proceeded across four discrete steps. First, we generated a complete set of descriptive statistics to identify outlying data points (i.e., those more than 3 *SD* from the group mean) and to test assumptions underlying subsequent inferential analyses. Our findings were such that we did not need to use nonparametric statistical comparisons. Second, a series of independent-sample *t*-tests (for continuous variables) and chi-square tests (for categorical variables) assessed between-group differences in key sociodemographic characteristics. Third, a series of independent-sample *t*-tests assessed between-group differences in sleep disturbances (as measured by GSAQ total score) and cognitive performance (as measured by the various BTACT outcome variables). Fourth, linear regression modelling investigated the possible mediating effect of sleep disruption on the relationship between presence of NFPA and cognitive dysfunction. Following

Baron and Kenny [49], we planned to run three separate models: the first describing the relationship between group status (patients versus controls) and cognitive performance, the second describing the relationship between group status and sleep disruption, and the third evaluating whether sleep disruption mediates the relationship between pituitary disease and cognitive dysfunction. We planned to thereafter use a Sobel test [50] to determine whether a model involving both pituitary disease and sleep disruption in predicting cognitive dysfunction would significantly stronger than that involving only pituitary disease as a predictor.

We also conducted these secondary analyses: (a) Pearson's product-moment correlation tests explored within-group associations between sleep disturbance and cognitive performance; (b) within the patient group only, Pearson's product-moment correlation tests investigated associations between sleep disturbance and clinical variables (age at diagnosis, disease duration, time since surgery and radiotherapy), and between cognitive performance and the same clinical variables; and (c) Mann-Whitney $U$ tests compared sleep disturbance and cognitive performance between patients who did and did not receive radiotherapy, and between those who did and did not have any comorbid medical conditions (i.e., diabetes, epilepsy, HIV, hydrocephalus, hypertension)

Given that our analyses featured multiple comparisons, we applied the Holm-Bonferroni correction factor and adjusted the P-value accordingly. Specifically, to correct for running 16 different analyses we set the significance level at p = 0.05/16 = 0.003. We also performed the bootstrapping method for analyses, using 1000 replicates.

### Ethics

Approval for the study procedures was granted by the Human Research Ethics Committees of the Department of Psychology (PSY 2020–029) and Faculty of Health Sciences Research and Ethics Committee (HREC REF: 462/2019) of the University of Cape Town. The study was performed in accordance with the ethical standards as laid down in the 1964 declaration of Helsinki and its later amendments or comparable ethical standards.

## Results

### Sample characteristics

The overall age range of the sample was 34–65 years ($M$ = 52.06, $SD$ = 9.79), with education ranging from 5–16 years ($M$ = 11.02, $SD$ = 2.50). As expected, given our case-control design and recruitment strategies, analyses detected no significant between-group differences for age, sex distribution, and education (see Table 1).

### Patient characteristics

Although age at NFPA diagnosis ranged quite widely, on average the diagnosis had been made more than 8 years prior to study enrollment. Most patients' pituitary tumours had been treated with both surgery and radiotherapy (see Table 2). At the time of the study, no patient was receiving radiotherapy and at least 1 year had elapsed since each patient's most recent radiotherapy treatment.

Seventeen of the 18 patients (94% of the sample) had been on a stable hormone replacement treatment regimen for at least 3 months prior to study enrolment; the remaining one did not require hormone replacement therapy. Replacement therapy included hydrocortisone ($n$ = 2; 11%), L-thyroxine ($n$ = 6; 33%), and a combination of the two ($n$ = 10; 56%). Two male patients had received Depo-Testosterone injections.

**Table 1. Sample sociodemographic characteristics (N = 37).**

| Variable | NFPA Patients (*n* = 18) | Healthy Controls (*n* = 19) | $t / \chi^2$ | P | *ESE* |
|---|---|---|---|---|---|
| Age (years) | | | | | |
| *M (SD)* | 52.44 (10.02) | 51.74 (9.29) | -0.21 | 0.83 | 0.07 |
| Range | 35–63 | 34–65 | | | |
| Education (years) | | | | | |
| *M (SD)* | 10.56 (2.04) | 11.68 (3.00) | 1.12 | 0.270 | 0.37 |
| Range | 4–15 | 5–16 | | | |
| Sex | | | 0.23 | 0.63 | 0.08 |
| Male (*f*, %) | 9 (50.00) | 11 (57.89) | | | |
| Female (*f*, %) | 9 (50.00) | 8 (42.11) | | | |

NFPA = non-functioning pituitary adenoma; ESE = effect size estimate—Cohen's d for t-tests, and Cramer's V for chi-square tests; M = mean; SD = standard deviation.

## Between-group comparisons: Sleep quality and cognition

On average, GSAQ scores for patients were slightly lower than those for controls. Analyses detected no significant between-group differences, however (see Table 3).

At the Bonferroni corrected level of significance, no between group differences were found for BTACT outcome measures. In contrast, analyses suggested there were significant between-group differences in cognitive performance on most individual subtests at the conventional level of significance (i.e, p < 0.05). On average, patients performed significantly more poorly (i.e., demonstrated significantly greater cognitive dysfunction) than controls on the BTACT overall score (adjusted and unadjusted for demographic variables except education) as well as on subtests assessing episodic memory and executive functioning (see Table 3).

**Table 2. NFPA patient characteristics (N = 18).**

| Variable | Statistic |
|---|---|
| Age at diagnosis (years) | |
| *M (SD)* | 45.5 (8.20) |
| Range | 30–57 |
| Duration of disease (years) | |
| *M (SD)* | 8.74 (4.38) |
| Range | 1.5–16 |
| Treatment | |
| Surgery only (*f*, %) | 5 (28%) |
| Surgery and radiotherapy (*f*, %) | 13 (72%) |
| Additional comorbidities (*f*, %) | |
| Smoking | 2 (10%) |
| Diabetes mellitus | 4 (20%) |
| Diabetes insipidus | 1 (5%) |
| Hypertension | 2 (10%) |
| BMI > 25 | 1 (5%) |

NFPA = non-functioning pituitary adenoma; M = mean; SD = standard deviation; BMI = body mass index.

**Table 3. Between-group comparisons: Sleep quality and cognitive performance (N = 37).**

| Outcome Variable | NFPA Patients (*n* = 18) | Controls (*n* = 19) | *t* | P | ESE | 95% CI LL | 95% CI UL |
|---|---|---|---|---|---|---|---|
| GSAQ (total score) | 36.17 (4.77) | 37.26 (3.97) | | 0.232 | 0.41 | -1.68 | 4.01 |
| BTACT | | | | | | | |
| Composite *z*-score [a] | -0.42 (0.39) | 0.00 (0.67) | | 0.012* | 0.75 | 0.05 | 0.75 |
| Episodic Memory | | | | | | | |
| Immediate recall | -0.56 (0.91) | 0.00 (1.00) | | 0.048* | 0.59 | -0.04 | 1.20 |
| Delayed recall | -0.61 (0.84) | 0.00 (1.00) | | 0.031* | 0.66 | 0.03 | 1.24 |
| Working Memory (Digit Span-Backwards) | -0.47 (0.84) | 0.00 (1.00) | | 0.067 | 0.51 | -0.09 | 1.09 |
| Executive Functioning | | | | | | | |
| Category fluency | -0.74 (0.65) | 0.00 (1.00) | | 0.005* | 0.87 | 0.17 | 1.24 |
| Stop and Go Task | -0.15 (0.75) | 0.22 (0.28) b | | 0.318 | 0.17 | -0.44 | 0.64 |
| Number Series | -0.66 (0.76) | 0.00 (1.00) | | 0.013* | 0.74 | 0.12 | 1.19 |
| Speed of Processing (Counting Backwards) | -0.18 (0.56) | 0.00 (1.00) | | 0.24 | 0.23 | -0.29 | 0.71 |
| Bi-factor *z*-score [e] | | | | | | | |
| Unadjusted | -0.90 (0.93) | -0.10 (1.10) | | 0.012* | 0.78 | 0.13 | 1.40 |
| Adjusted for: | | | | | | | |
| Age | -1.31 (1.15) | -0.35 (1.33) | | 0.013* | 0.77 | 0.17 | 1.71 |
| Education | -0.05 (1.21) | 0.56 (1.31) | | 0.077 | 0.48 | -0.22 | 1.41 |
| Gender | -1.06 (1.05) | -0.14 (1.26) | | 0.011* | 0.79 | 0.15 | 1.62 |
| Age, education | -0.41 (1.27) | 0.31 (1.38) | | 0.051 | 0.54 | -0.17 | 1.56 |
| Age, gender | -1.30 (1.13) | -0.33 (1.34) | | 0.012* | 0.78 | 0.19 | 1.71 |
| Age, education, gender | -0.39 (1.24) | 0.35 (1.39) | | 0.045* | 0.56 | -0.14 | 1.56 |

In columns 2 and 3, means are presented with standard deviations in parentheses. P-value considered statistically significant at the Bonferroni corrected p < 0.003.

NFPA = non-functioning pituitary adenoma; ESE = effect size estimate (Cohen's *d*); CI = confidence interval; LL = lower limit; UL = upper limit; GSAQ = Global Sleep Assessment Questionnaire; BTACT = Brief Test of Adult Cognition by Telephone.

[a] Conventional scoring method for the BTACT. *n* = 18; data from one participant in this group were omitted from analysis because they contained outlying values.

[e] Bi-factor scoring method for the BTACT.

*Statistically significant at the conventional level (p < 0.05).

## Mediation testing: Regression analyses

Linear regression modeling indicated that group status was not a significant predictor of cognitive performance at the Bonferroni corrected significance level (see Table 4). However, at the conventional level of significance group status was a significant predictor of cognitive performance. This significant relationship held for several individual BTACT subtests (Episodic Memory [both immediate and delayed recall], Category Fluency, Number Series) and for overall BTACT performance (both adjusted and unadjusted for demographic variables except those including education).

Similar modelling found that group status did not significantly predict sleep quality (see Table 4), thus indicating an absence of the mediating effect of sleep on the relationship between the presence of NFPA and cognitive dysfunction.

## Secondary analyses

Within-group bivariate correlational analyses detected no significant associations between sleep disturbance and cognitive performance at the Bonferroni corrected significance. However, at the conventional level of significance, one significant association was found. In the

**Table 4. Regression analysis: Group status (NFPA patients versus healthy controls) as a predictor of sleep quality and cognitive performance (N = 37).**

| Outcome Variable | ß | SE | 95% CI LL | 95% CI UL | p | $R^2$ |
|---|---|---|---|---|---|---|
| GSAQ (total score) | -1.10 | 1.43 | -3.90 | 1.57 | 0.232 | -0.01 |
| BTACT | | | | | | |
| Composite z-score [a] | -0.42 | 0.18 | -0.75 | -0.04 | 0.020* | 0.10 |
| Episodic Memory | | | | | | |
| Immediate recall | -0.56 | 0.31 | -1.19 | -0.06 | 0.049* | 0.06 |
| Delayed recall | -0.61 | 0.30 | -1.21 | -0.02 | 0.026* | 0.07 |
| Working Memory | -0.47 | 0.29 | -1.05 | 0.13 | 0.068 | 0.04 |
| Digit Span-Backwards | | | | | | |
| Executive Functioning | | | | | | |
| Category fluency | -0.74 | 0.27 | -1.27 | -0.19 | 0.009* | 0.14 |
| Stop and Go Task [b] | -0.15 | 0.28 | -0.68 | 0.43 | 0.307 | 0.01 |
| Number Series | -0.66 | 0.30 | -1.23 | -0.07 | 0.023* | 0.10 |
| Speed of Processing | | | | | | |
| Counting Backwards | -0.18 | 0.26 | -0.71 | 0.32 | 0.235 | -0.015 |
| Bi-factor z-score c | | | | | | |
| Unadjusted | -0.79 | 0.2 | -1.41 | -0.13 | 0.014* | 0.11 |
| Adjusted for: | | | | | | |
| Age | -0.96 | 0.39 | -1.71 | -0.17 | 0.016* | 0.11 |
| Education | -0.61 | 0.41 | -1.44 | 0.15 | 0.074 | 0.03 |
| Gender | -0.91 | 0.38 | -1.64 | -0.14 | 0.011* | 0.12 |
| Age, education | -0.72 | 0.43 | -1.54 | 0.10 | 0.057 | 0.05 |
| Age, gender | -0.97 | 0.41 | -1.76 | -0.16 | 0.013* | 0.12 |
| Age, education, gender | -0.74 | 0.42 | -1.49 | 0.11 | 0.047* | 0.05 |

Degrees of freedom for *F*-statistic are (1, 35). P-value considered statistically significant at the Bonferroni corrected p < 0.003. NFPA = non-functioning pituitary adenoma; GSAQ = Global Sleep Assessment Questionnaire; BTACT = Brief Test of Adult Cognition by Telephone; CI = confidence interval; LL = lower limit; UL = upper limit. For the regression analyses, the Control Group was the reference group.

[a] Conventional scoring method for the BTACT.

[b] N = 36; data from one control participant were omitted from analysis because they contained outlying values. Bi-factor scoring method for the BTACT.

*Statistically significant at the conventional level (p < 0.05).

patient group, GSAQ score and Stop and Go Task total score were significantly positively correlated (i.e., patients with less sleep disturbance performed better on this subtest; see Table 5). Despite the overall lack of significance in these secondary analyses, it is worth noting that there are stronger and more positive correlations present in the control group. This pattern suggests that, in healthy adults, less sleep disturbance is more likely to be associated with better cognitive performance than it is in NFPA patients. However, the differences failed to reach statistical significance.

Bivariate correlational analyses of data from the patient group detected no significant associations between sleep disturbance and age at diagnosis, disease duration, and time since surgery and radiotherapy, or between cognitive performance and the same set of clinical variables (all ps > 0.104).

Mann-Whitney *U* tests detected significant performance differences (at the conventional level of significance) between patients who did (*n* = 12) and did not (*n* = 5) receive radiotherapy on two BTACT subtests (Working Memory, p = 0.041; and Recall, p = 0.041;) and on the BTACT bi-factor z-score adjusted for education (p = 0.041) (see S1, S2 Figs). In each of these

**Table 5. Bivariate correlational analyses: Relationship between GSAQ score and BTACT performance within each group (N = 37).**

| BTACT Outcome Variable | NFPA Patients (*n* = 18) | | Healthy Controls (*n* = 19) | | |
|---|---|---|---|---|---|
| | *r* | P | *r* | p | *p* |
| Composite *z*-score [a] | 0.14 | 0.58 | 0.11 | 0.65 | 0.47 |
| Episodic Memory | | | | | |
| Immediate recall | 0.21 | 0.41 | 0.31 | 0.20 | 0.38 |
| Delayed recall | 0.03 | 0.89 | 0.14 | 0.57 | 0.38 |
| Working Memory | | | | | |
| Digit Span-Backwards | -0.17 | 0.51 | 0.14 | 0.56 | 0.19 |
| Executive Functioning | | | | | |
| Category Fluency | 0.02 | 0.94 | 0.29 | 0.24 | 0.22 |
| Stop and Go Task | 0.48 | 0.04* | 0.43 b | 0.08 | 0.43 |
| Number Series | 0.29 | 0.25 | -0.13 | 0.59 | 0.17 |
| Speed of Processing | | | | | |
| Counting Backwards | -0.02 | 0.93 | 0.05 | 0.83 | 0.42 |

Data presented are Pearson's correlation coefficients (*r*) and associated *p*-values representing magnitude of association (within each study group) between the listed cognitive outcome variable and GSAQ score, which measures sleep quality. P-value considered statistically significant at the Bonferroni corrected p < 0.003.
GSAQ = Global Sleep Assessment Questionnaire; BTACT = Brief Test of Adult Cognition by Telephone; NFPA = non-functioning pituitary adenoma.
[a] Conventional scoring method for the BTACT.
[b] *n* = 18; data from one participant in this group were omitted from analysis because they contained outlying values.
*Statistically significant at the conventional level (p < 0.05).

cases, the order of the means suggested that patients who had received radiotherapy performed significantly more poorly than those who had not.

 Mann-Whitney *U* tests detected no significant differences in cognitive performance between patients who did (*n* = 10) and did not (*n* = 10) have medical comorbidities (all ps > 0.102).

## Discussion

The primary objective of this study was to examine sleep quality and cognitive functioning in a sample of patients with non-functioning pituitary adenomas (NFPA), with an ultimate aim of determining whether sleep disruption mediates cognitive dysfunction in those patients. We recruited 37 participants (18 patients and 19 socio-demographically matched healthy controls), took objective assessments of their cognitive functioning, and asked for self-reports regarding sleep quality.

 Our first hypothesis was that patient performance on a standardized test of cognitive performance (the Brief Test of Adult Cognition by Telephone [BTACT]; [45]] would be significantly worse than that of controls. Analyses did not confirm this prediction. Although patients scored more poorly on both the global BTACT index and on most individual BTACT subtests, including those assessing episodic memory and various aspects of executive functioning, these failed to reach statistical significance.

 This result may not be surprising–a meta-analysis by Pertichetti et al. [51] found that altered memory was only observed in 22% of NFPA cases, compared to higher rates in

Cushing's disease and Acromegaly. This meta-analysis showed that patients with NFPA had greatest difficulties in mental agility and memory recall. The BTACT, although a valid test of cognitive functioning, does not contain tests of mental agility, and only one memory test. Perhaps the BTACT is not sensitive enough to detect deficits in patients with NFPA. Another explanation for our non-significant findings could be related to treatment. Hendrix et al. [52] showed that neurocognitive impairments resolved within 2 months after surgery in patients with NFPA. Our patient group had an average disease duration of 8.74 years, indicating that time since surgery had been a lot longer than 2 months. This may explain the non-significant findings in neurocognitive impairment. In addition, the type of surgery seems important, with some studies showing improvement and others worsening cognitive functioning depending on the type of surgery performed [51]. In our sample we did not collect information on the type of surgery, which may play an important role in cognitive functioning post-surgery.

Although our data did not reach statistical significance at the Bonferroni corrected level, we found that patients performed significantly worse than controls at the conventional level of significance. These data patterns are consistent with reports from most previously published studies in this area. Although there is not a particularly large body of evidence, several studies report that patients with pituitary disease (including those with NFPA, and regardless of whether treatment has included surgery only or surgery plus radiotherapy) present with a variety of cognitive difficulties, including problems with visuoconstructional abilities, learning, memory, and executive functioning [28, 29].

Several different mechanisms could account for the manifestation of cognitive difficulties in patients with pituitary disease. Naturally, the primary disease has effects on brain structure and function, but hormonal changes secondary to the tumour and effects of treatment interventions may play a role as well. In our sample, 12 of the 18 patients with NFPA were on glucocorticoid replacement therapy. A large body of research indicates that cognitive performance (particularly in the domains of memory and executive functioning) is affected by fluctuating glucocorticoid levels, with impairment noted when those levels either exceed or dip below physiologically normal ranges [53]. Similarly, 16 of our 18 patients were on thyroxine replacement, and thyroid hormonal dysregulation can affect both cognitive and behavioral functioning [54]. Our small sample size may have accounted for us only finding significant differences at the conventional level of significance. As such, studies with larger samples should investigate cognitive functioning in NFPA's.

Previous research also suggests that performance in the cognitive domains of memory and executive functioning are particularly susceptible to the effects of surgical and radiotherapy treatment for pituitary disease [28]. For instance, Brummelman et al. [27] found that NFPA patients who had experienced both surgical and radiotherapy treatment performed more poorly on tests of learning and memory than patients who had undergone surgery only. Similarly, Noad et al. [30] found that patients with pituitary tumours who had experienced both surgical and radiotherapy treatment performed significantly more poorly on tests of executive functioning than patients who had undergone surgery only.

An implication of these findings, naturally, is that radiotherapy has a much more negative effect on cognition than does surgery. Results from one of our secondary analyses are consistent with that inference: We found that performance on tests of working memory and episodic memory, in particular, was significantly worse (at the conventional level of significance) in patients who had received radiotherapy than in those who had not.

Our second hypothesis was that patients would self-report significantly more disrupted sleep than controls on the Global Sleep Assessment Questionnaire (GSAQ). Analyses did not confirm this prediction. Our GSAQ data were fairly homogenous across groups, with few participants reporting any significant issues with sleep quality.

These results contrast with data from previous studies in which patients with pituitary disease (including NFPA) reported poorer sleep quality and more sleep disturbances [26]. Although it is possible that the reason for the currently observed outcome is that there were truly no between-group differences in sleep quality, one must consider the context established by previous studies and seek alternative reasons for the non-significant finding.

One such reason may be the instrument used to assess sleep quality. The GSAQ is a self-report instrument, with questions pertaining to the respondent's subjective experiences of sleep. Hence, it does not provide data on objective sleep characteristics, such as duration or density of each sleep stage. Previous studies investigating the effects of pituitary disease on sleep quality often measure those characteristics, and subsequently report that patients' sleep is objectively disrupted [24, 25, 31, 55]. Hence, one may speculate that the self-report GSAQ is a crude measure of sleep disturbance and may fail to elicit finer, objectively measurable, aberrations of sleep architecture and quality.

Such speculation is supported in two ways. First, objective and subjective measures of sleep quality are often weakly related. For instance, certain sleep parameters, such as duration or onset latency, are often overestimated by subjective measures in comparison to objective measures such as actigraphy or polysomnography (PSG) [56, 57]. Lauderdale et al. [58] argued that self-report measures of sleep are often inaccurate because they ask respondents to integrate information regarding sleep duration and quality from several days (and sometimes several weeks) when, in reality, sleep varies from night to night. In one particularly notable study, Lipinska and Thomas [59] found that, in their sample of 21 women with posttraumatic stress disorder (PTSD), 19 women with trauma exposure but no PTSD, and 20 healthy controls, there were few significant correlations between self-reported sleep quality averaged over the previous 30 days and PSG sleep data collected in the laboratory.

Second, although the GSAQ may be proficient at detecting clinical sleep disorders and distinguishing between them [43], it may not be sensitive to milder (i.e., sub-clinical) forms of sleep disruption or to generally poor sleep quality. Patients in our study may have been experiencing minimal sleep disruption and/or poor sleep quality rather than a clinically diagnosable sleep disorder.

Third, it is possible that patients with pituitary adenoma may normalize their sleep disturbances over time and hence may fail to recognize that they do indeed experience impaired sleep quality. In short, assessment of sleep quality using a self-report measure has significant shortcomings and it might have been preferable to assess this construct using objective measures.

Our third hypothesis stated that sleep disturbance may mediate the relationship between the presence of pituitary disease and poor cognitive performance. Linear regression models confirmed a significant predictive relationship between group status (patients versus controls) and cognitive performance but detected no significant predictive relationship between group membership and sleep disturbance. Hence, this hypothesis was not confirmed.

This non-significant association between sleep and cognition contrasts with previous literature suggesting that healthy sleep promotes intact cognitive performance [16, 60, 61]. This discrepancy between our findings and those described by previously published studies may, as we note above, be a consequence of the GSAQ's limited ability to report accurately on sleep quality and its lack of sensitivity in detecting sub-clinical sleep disturbances. Despite the lack of significance observed here, it is still worthwhile for future studies to pursue the potential mediational role that sleep may play in the relationship between pituitary disease and cognitive dysfunction: If this third variable is determined to have a significant impact on cognitive functioning in these patients, then sleep interventions can be targeted as a viable means of improving their quality of life.

## Limitations

The following limitations might constrain the current study's inferential reach.

First, the study's sample size ($N = 37$) means it may not have had adequate statistical power to achieve its aims. G*Power software [62] suggested that a minimum sample size be set at $N = 62$ (31 per group) to run a mediation analysis with one independent variable and one mediator if the expected effect size is of medium magnitude (Cohen's $f^2 = 0.15$) and the desired statistical power is at least 0.85. Because only one other study [31] examined both sleep disruption and cognitive dysfunction in patients with pituitary disease, we estimated a medium effect size based on the well-known relationship between the two [29, 63]. Naturally, if the effect size in the population was smaller, an even larger sample would be needed in order to power the investigation adequately. Hence, future studies in this area should aim to recruit substantially larger groups of patients and controls. However, it should be noted that pituitary disease occurs quite rarely, with prevalence rates estimated as being 7–41.3 / 100 000 population [64]. Given this rarity, the number of people available to act as participants in research is small, making recruiting statistically powerful sample sizes a difficult task.

Second, because we excluded individuals with self-reported psychiatric illness from participation, we could not consider the relationship between pituitary disease and affective disorders. There is evidence suggesting that the hormonal imbalances occurring in patients with NFPA may result in symptoms of depression [65]. A separate body of literature describes the spectrum of cognitive deficits observed in depressed individuals [66]. The cognitive dysfunction observed in pituitary disease patients may, therefore, be explained at least partially by the presence of psychiatric illness.

Third, we also did not consider the effects of hormone replacement therapy on cognitive functioning. Although numerous studies of hydrocortisone replacement therapy suggest that neither dosage nor length of treatment influence cognitive test performance [67, 68], it might still have been worthwhile examining the individual treatment regimens of the patient sample. (A related note here is that, because growth hormone is not freely available in the South African clinical context, the absence of this avenue of treatment for most patients in this sample may have indirectly impeded peak cognitive performance [69]). Other potential confounding variables not considered in the current design were time of day cognitive tests were administered relative to hydrocortisone dose (it is possible, for instance, that some patients were experiencing low cortisol concentrations at the time of testing and hence did not deliver their best possible BTACT performance) and dose of radiation therapy experienced.

Fourth, there are limitations related to the mode and form of cognitive assessment we used. Although administration of the BTACT (and similar telephone-based or online assessments) is valuable in ensuring participants can be reached easily and conveniently, such administration (a) offers relatively little control over potential distractors such as external noise and the presence of others in the room during assessment, (b) might be more negatively affected by participants' hearing impairment than might in-person assessment administration, (c) is limited to the verbal modality only, meaning cognitive domains such as visuospatial memory and psychomotor coordination remain unassessed. Additionally, it might be argued that interpretation of the participants' cognitive performance is affected by the lack of information about their overall IQ and by the fact that the cognitive tasks used in the study were of low ecological validity [70].

Finally, our patient group comprised exclusively those with NFPA. Hence, we cannot comment on relations between disease presence, sleep quality, and cognition in functional pituitary diseases. Exploring those associations in patients with those conditions might be valuable

given that sleep dysfunction and cognitive impairment are known to exist in acromegaly and Cushing's disease [29, 71].

## Summary and conclusion

We observed poorer cognitive performance in NFPA patients than in sociodemographically matched healthy controls, although these differences failed to reach statistical significance at the Bonferroni corrected level of significance. Nevertheless, these results suggests that health professionals treating these patients must have a keen awareness of the possibility that cognitive difficulties may impact on treatment adherence and outcomes. Further, it may be important for those professionals to enquire from significant others about possible cognitive impairment, given that patients themselves may not have insight into their difficulties.

Our analyses detected no significant between-group differences in sleep disturbance and no significant association between sleep disturbance and cognitive dysfunction. Given that we did not measure sleep quality objectively, that the nuances of sleep architecture cannot be subjectively reported on, and that cognitive impairment in medically ill patients is often associated with disease-related sleep disruption, further research should be conducted to establish whether sleep disruption does indeed serve as a mediating variable in the relationship between NFPA and cognitive dysfunction.

## Supporting information

**S1 Fig. Between-group differences on BTACT outcome variables: Radiotherapy versus no radiotherapy.**
(TIF)

**S2 Fig. Between-group differences on BTACT Bi-factor outcome variables: Radiotherapy versus no radiotherapy.**
(TIF)

## Acknowledgments

We acknowledge the staff at Groote Schuur Hospital, for allowing us to use their facilities and resources during data collection.

## Author Contributions

**Conceptualization:** Olivia de Villiers, Claudia Elliot-Wilson, Kevin G. F. Thomas, Thurandrie Naiker, Michelle Henry, Ian L. Ross.

**Data curation:** Olivia de Villiers, Claudia Elliot-Wilson.

**Formal analysis:** Michelle Henry.

**Investigation:** Kevin G. F. Thomas, Ian L. Ross.

**Methodology:** Kevin G. F. Thomas, Michelle Henry, Ian L. Ross.

**Project administration:** Kevin G. F. Thomas, Ian L. Ross.

**Supervision:** Kevin G. F. Thomas, Patrick L. Semple, Thurandrie Naiker, Michelle Henry.

**Writing – original draft:** Olivia de Villiers, Claudia Elliot-Wilson, Kevin G. F. Thomas, Michelle Henry, Ian L. Ross.

**Writing – review & editing:** Kevin G. F. Thomas, Patrick L. Semple, Thurandrie Naiker, Michelle Henry, Ian L. Ross.

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
