## [Decision Letter · Decision Letter 0]

19 Jul 2023

PONE-D-23-02515Sleep and Cognition in South African Patients with Non-Functioning Pituitary AdenomasPLOS ONE

Dear Dr. Henry,

Thank you for submitting your manuscript to PLOS ONE. I have now received reviews for your manuscript from three reviewers who have expertise in the area you are investigating.

The reviewers’ comments are provided at the end of this letter. As you will see when you read their critiques, reviewers feel that the manuscript focuses on an interesting research topic. However, all reviewers raised a number major of concerns that should be addressed in order for the paper to be suitable for publication.

Specifically, all reviewers call for more clarifications to be provided in several parts of the manuscript – most notably, when outlining the formulation of the hypotheses in the Introduction, key aspects of the Method section, statistical analyses, as well as certain conclusions presented in the Abstract and the Discussion. A major point has to do with the process followed during the selection of the participants in the patient group and the inclusion / exclusion criteria, because several individuals appear to have other major health issues that may in fact affect cognitive functioning. Moreover, given the very small sample size, the decision to split the patient group into two subgroups may create problematic issues with the statistical analysis. Therefore, a power analysis for each separate statistical analysis should also be provided to establish whether the sample size was sufficient for at least a medium effect size to be observed.

Addressing these comments would strengthen the clarity of the manuscript and increase its potential impact. Therefore, I invite you to submit a revised version of the manuscript that addresses the points raised during the review process.

We look forward to receiving your revised manuscript.

Kind regards,

Ioanna Markostamou, Ph.D.

Academic Editor

PLOS ONE

Journal Requirements: 

Reviewers' comments:

Reviewer's Responses to Questions

**Comments to the Author**

1. Is the manuscript technically sound, and do the data support the conclusions?

Reviewer #1: Partly

Reviewer #2: Yes

Reviewer #3: Yes

2. Has the statistical analysis been performed appropriately and rigorously? 

Reviewer #1: No

Reviewer #2: Yes

Reviewer #3: I Don't Know

3. Have the authors made all data underlying the findings in their manuscript fully available?

Reviewer #1: Yes

Reviewer #2: Yes

Reviewer #3: No

4. Is the manuscript presented in an intelligible fashion and written in standard English?

Reviewer #1: Yes

Reviewer #2: Yes

Reviewer #3: Yes

5. Review Comments to the Author

Reviewer #1: Overall, this is an interesting study and has the potential to shed more light on the relationship between sleep and cognitive functions in patients with non-functioning pituitary. The statistical analysis is generally appropriate and the results are well presented but some changes can be considered. The sample size is very low. The introduction and discussion are suitable but some issues should be discussed. In my opinion, there are some major and minor points that should be considered.

Major points:

1. In the introduction section, there is no need to provide statistical results from other studies. It should be modified.

2. In my opinion, there is a lack of general justification for the hypotheses. I appreciate that authors describe the limitations of the Wennberg et al. (2019) study. However, they should provide more general reasons for the aims of their study. Perhaps based on the gaps from other studies and/or neuropsychological mechanisms of importance pituitary role for cognition.

3. In my opinion, two patients may have cognitive dysfunctions as an impact of other diseases (HIV and epilepsy). I suggest to consider remove them and conduct a new statistical analysis.

Suggested references:

Woods, S. P., Moore, D. J., Weber, E., & Grant, I. (2009). Cognitive neuropsychology of HIV-associated neurocognitive disorders. Neuropsychology Review, 19, 152-168.

McCagh, J., Fisk, J. E., & Baker, G. A. (2009). Epilepsy, psychosocial and cognitive functioning. Epilepsy Research, 86, 1-14.

4. Did authors check the normal distribution of the analysis variables (S-W test, skewness, and kurtosis).

5. I strongly suggest using correction for multiple comparisons (e.g., Holm-Bonferroni or another) and adding a p-value after correction to the results.

6. I suggest considering using a bootstrapping method to improve the statistical analysis of comparisons and relationships.

7. I strongly suggest presenting some of the results in the Figures as the box or violin plots (e.g., for comparison analysis).

8. In the discussion section authors should add propositions for clinical practice.

9. In my opinion, there are much more limitations of this study. First, the use of neuropsychological tests during a phone call raises controversy, e.g., lack of control over potential distractors (noise, room conditions, and presence of other people during the test) or hearing impairments in the subjects. It is the main limitation of this study and authors should discuss it. Second, there is no information about the IQ of the participants and any methods to measure it. Third, only verbal forms of neuropsychological tests were used in the study. So, we can't interpret the motor aspects of cognitive functioning e.g., cognitive flexibility or motor control. It should be discussed. Finally, tasks used in this study are characterized by low "ecological validity". Please discuss the excellent work by Chaytor & Schmitter-Edgecombe (2003).

Suggested reference:

Chaytor, N., & Schmitter-Edgecombe, M. (2003). The ecological validity of neuropsychological tests: A review of the literature on everyday cognitive skills. Neuropsychology Review, 13, 181-197.

Minor points:

1. The text should be checked for mistakes (e.g., in the text and Tables "p" should be written in lowercase).

2. There is no need to use stars when authors provide a p-value.

3. Is ESE the value of Cohen's d? It should be explained.

Reviewer #2: In this study, 20 patients with non-functioning pituitary adenomas and 19 healthy controls were examined. The patients demonstrated significant cognitive impairment across various domains, but there were no significant differences in sleep between the groups, and sleep disruption did not correlate with cognitive performance. It's a relatively understudied topic, and it's interesting. However, there are still some issues that need to be clarified.

1. Please add detailed inclusion and exclusion criteria in the Methods section. The author mentioned excluding people with previous cognitive impairment, how was cognitive impairment identified here?

2. On page 7, line 173, the author mentioned that people with diseases that may affect cognitive function and sleep quality were excluded. Please describe in detail which diseases were included.

3. On page 12, line 282, please add the full name when the abbreviation appears for the first time, such as M, SD. At the same time, the full names of M and SD should be added to the notes in Table 1 and Table 2.

4. In regression models with the GSAQ score and multiple BTACT scores as outcomes, were other covariates considered? Age, education, and gender appeared to be considered in the model with the Bi-factor scoring method for the BTACT outcome.

Reviewer #3: In the manuscript, Villiers and colleagues investigated whether sleep and cognition were disrupted in patients with non-functioning pituitary adenomas as compared to healthy controls. Participants were interviewed over the phone and completed the Global Sleep Assessment Questionnaire and Brief Test of Adult Cognition, thus providing measures of self-reported sleep and cognitive performance, respectively. The authors found evidence of cognitive disruption, but no significant difference in their sleep measure. Given the limited research on the link between sleep and cognition in patients with pituitary disease, the authors’ question is interesting and relevant. However, I have some concerns that I would recommend addressing before publishing the manuscript.

Major points:

- Overall, a series of statistical comparisons and correlations have been made, but it is unclear whether the authors corrected for multiple comparisons. If not, the type I error (false positive) rate can be inflated possibly leading to erroneous inferences. I would strongly recommend correcting for multiple comparisons using an appropriate method and reporting transparently when the results are corrected/uncorrected (e.g., including this in the tables).

- Furthermore, the rationale for the median split analyses separating the participants into ‘good’ and ‘bad’ sleepers is unclear, especially given the low sample size (N=10 and N=9). This could also contribute to the multiple comparisons problem. Please provide a clear rationale for these analyses, and report whether the results are corrected for multiple comparisons.

- Given that subjective and objective sleep is often not correlated, this study may not have been sufficiently sensitive to capture sleep disruption. The authors acknowledge this in the discussion. However, the abstract is rather misleading in saying that the patients’ sleep quality is intact. In the absence of evidence for the null (e.g., equivalence testing, Bayesian, etc.), I would recommend modifying the abstract to better reflect their conclusions drawn in the main text. Furthermore, considering that the patients have been living with pituitary adenoma for several years, their perception of what is ‘a good night of sleep’ will likely be different to that of healthy controls. Consider adding this point to the discussion.

- In the results section under ‘secondary analyses’, the authors infer that less sleep disturbance is likely associated with better cognitive performance than in the patients, however, the results are not significant. Did the authors compare the correlations? If not, please change the language accordingly to reflect a null result.

- I was unable to find the dataset using the link provided and when searching for it on ZivaHub. Please provide a correct link.

Minor points:

- The page numbers are not accurate, please correct.

- Page 3, lines 71-72: “… characterized by deep sleep and slow wave sleep (SWS on electroencephalogram readings”. I presume the authors meant that stage 3 is characterized by deep sleep or slow wave sleep. Also, the stages are characterized by polysomnography, usually meaning EEG as well as EOG and EMG.

- Page 4, middle paragraph: With the concluding sentence suggesting that sleep is important for consolidation on subsequent days, it is unclear how this relates to the present study, considering that there was only one measure of episodic memory on the same day. Consider providing a clearer link to the present question.

- Page 6, first paragraph: When introducing the previous similar study, it would be helpful to include which measures of sleep and cognition were administered for ease of comparison. This could also be reiterated when discussing the limitations of the sleep measure in the discussion.

6. PLOS authors have the option to publish the peer review history of their article (what does this mean?). If published, this will include your full peer review and any attached files.

Reviewer #1: No

Reviewer #2: No

Reviewer #3: No

---

## [Author Response · Author response to Decision Letter 0]

12 Oct 2023

We have responded to the editor and reviewers individually in our uploaded letter.

---

## [Decision Letter · Decision Letter 1]

14 Nov 2023

PONE-D-23-02515R1Sleep and Cognition in South African Patients with Non-Functioning Pituitary AdenomasPLOS ONE

Dear Dr. Henry,

Thank you for submitting your revised manuscript to PLOS ONE. I have now received the comments on your revised manuscript from two of the original reviewers. While Reviewer 2 feels that the issues they had previously raised were fully addressed in the revision, Reviewer 3 outlines several important issues that have not been appropriately addressed. Therefore, I would like to invite you to submit a revised version of the manuscript that addresses all points raised during the review process. Please find the reviewers' comments at the end of this letter.

We look forward to receiving your revised manuscript.

Kind regards,

Ioanna Markostamou, Ph.D.

Academic Editor

PLOS ONE

Reviewers' comments:

Reviewer's Responses to Questions

**Comments to the Author**

1. If the authors have adequately addressed your comments raised in a previous round of review and you feel that this manuscript is now acceptable for publication, you may indicate that here to bypass the “Comments to the Author” section, enter your conflict of interest statement in the “Confidential to Editor” section, and submit your "Accept" recommendation.

Reviewer #2: All comments have been addressed

Reviewer #3: (No Response)

2. Is the manuscript technically sound, and do the data support the conclusions?

Reviewer #2: (No Response)

Reviewer #3: Partly

3. Has the statistical analysis been performed appropriately and rigorously? 

Reviewer #2: (No Response)

Reviewer #3: Yes

4. Have the authors made all data underlying the findings in their manuscript fully available?

Reviewer #2: (No Response)

Reviewer #3: Yes

5. Is the manuscript presented in an intelligible fashion and written in standard English?

Reviewer #2: (No Response)

Reviewer #3: Yes

6. Review Comments to the Author

Reviewer #2: (No Response)

Reviewer #3: I thank the Authors for their work on the revisions and for addressing my concerns to some extent. Where these have not been fully addressed, I have included the original point, the Author response, and what, in my opinion, remains to be addressed. I believe these outstanding concerns could be addressed with some revisions.

Major points

1. My original concern:

Overall, a series of statistical comparisons and correlations have been made, but it is unclear whether the authors corrected for multiple comparisons. If not, the type I error (false positive) rate can be inflated possibly leading to erroneous inferences. I would strongly recommend correcting for multiple comparisons using an appropriate method and reporting transparently when the results are corrected/uncorrected (e.g., including this in the tables).

Author response:

Please see our response to Reviewer 1’s Comment 5.

Updated concern:

While the Authors have addressed my concern and corrected for multiple comparisons, this has had no bearing on their interpretation of the results in the abstract, discussion, and conclusion. After applying their correction methods, they did not find any significant differences, however, this is not reflected in the overall tone and take-home messages of the paper. I would recommend highlighting this in the relevant sections (abstract and conclusion).

In the table notes, there is no mention of whether the p-values are corrected for multiple comparisons or not. Please include this on all table notes.

2. My original concern:

Furthermore, the rationale for the median split analyses separating the participants into ‘good’ and ‘bad’ sleepers is unclear, especially given the low sample size (N=10 and N=9). This could also contribute to the multiple comparisons problem. Please provide a clear rationale for these analyses, and report whether the results are corrected for multiple comparisons.

Author response:

We thank the reviewer for this question. We divided patients into good and bad sleepers based on the cut-offs recommended by developers of the Global Sleep Assessment Questionnaires (GSAQ). We also, as noted above, corrected for multiple comparisons using the Holm-Bonferroni correction factor. We now include the following statement in the Methods section:

We delineated patients into good and bad sleepers, based on the control groups median GSAQ score since there are no recommended cut-offs for the GSQA.

Updated concern:

My concern was not with the choice of cut-off threshold, but rather with why they separated the group in the first place, i.e., what was the rationale that made them split the group into good and bad sleepers. Considering the small sample size, I think there needs to be a solid reason for this analysis.

Minor points

My original concern:

Page 4, middle paragraph: With the concluding sentence suggesting that sleep is important for consolidation on subsequent days, it is unclear how this relates to the present study, considering that there was only one measure of episodic memory on the same day. Consider providing a clearer link to the present question.

Author response:

Memory consolidation processes occurring during sleep may be similar to those occurring during waking (see, e.g., Siegel, 2021; Wamsley, 2019), although as reflected by our review there are many more studies of sleep-dependent consolidation than waking consolidation. Nonetheless, we take the Reviewer’s point that our exclusive citing of literature on sleep-dependent consolidation is not, at first glance, directly related to the design of the current study. We have added some text to the relevant part of the Introduction in an attempt to clarify the links between our study and the literature on memory consolidation.

Updated concern:

While I appreciate that the Authors want to draw a parallel between the literature on memory consolidation during wake and sleep, these two paragraphs (on memory consolidation during sleep and wake) seem to make the rationale for looking at sleep disruption and memory even more confusing. Apologies if I was unclear when I recommended that the Authors make the rationale clearer. The key discrepancy between the literature they are referring to and their study is that this study is not measuring memory across a period of sleep, but rather whether poor sleep is linked to poor memory formation. What might help the rationale is to focus less on sleep-associated memory consolidation and instead focus on the literature which has found that sleep disruption prior to learning is detrimental for memory formation (for example, see this meta-analysis: https://doi.org/10.1037/bul0000348). The Authors have included a sentence on this, but the take-home message of that paragraph seems to be that sleep disruption after learning is detrimental for consolidation.

7. PLOS authors have the option to publish the peer review history of their article (what does this mean?). If published, this will include your full peer review and any attached files.

Reviewer #2: No

Reviewer #3: No

---

## [Author Response · Author response to Decision Letter 1]

6 Dec 2023

06/12/2023

The Editor

PLoS one

Regarding: Sleep and cognition in South African patients with non-functioning pituitary adenomas

On behalf of my co-authors we thank you most sincerely for the review and suggestions, which we feel will improve the manuscript considerably. We have adopted all the suggestions and we believe that the manuscript is now ready for publication.

Reviewer Comments

Major points

Updated concern:

While the Authors have addressed my concern and corrected for multiple comparisons, this has had no bearing on their interpretation of the results in the abstract, discussion, and conclusion. After applying their correction methods, they did not find any significant differences, however, this is not reflected in the overall tone and take-home messages of the paper. I would recommend highlighting this in the relevant sections (abstract and conclusion).

Updated Response: We have edited the abstract, discussion and conclusion to better reflect the results.

We have added the following sentence to the abstract:

We found no significant differences in cognition between patients and controls after adjusting for multiple comparisons.

We have changed the sentence in the abstract to read:

Our data suggest that NFPA patients’ cognition and sleep quality is relatively intact, and that sleep disruption does mediate the cognitive

We have also made multiple edits to the discussion and conclusion to better reflect our non-significant findings (see tracked changes manuscript). 

Updated concern: In the table notes, there is no mention of whether the p-values are corrected for multiple comparisons or not. Please include this on all table notes.

Response: P-values were not corrected for multiple comparisons, but were only considered significant if p was less than 0.003 to allow for multiple comparisons. We have mentioned this in the statistical analyses section of the paper and added this note to Tables 3-5. 

Updated concern:

My concern was not with the choice of cut-off threshold, but rather with why they separated the group in the first place, i.e., what was the rationale that made them split the group into good and bad sleepers. Considering the small sample size, I think there needs to be a solid reason for this analysis.

Response: We have removed this analysis from the paper. 

Minor points

Updated concern:

While I appreciate that the Authors want to draw a parallel between the literature on memory consolidation during wake and sleep, these two paragraphs (on memory consolidation during sleep and wake) seem to make the rationale for looking at sleep disruption and memory even more confusing. Apologies if I was unclear when I recommended that the Authors make the rationale clearer. The key discrepancy between the literature they are referring to and their study is that this study is not measuring memory across a period of sleep, but rather whether poor sleep is linked to poor memory formation. What might help the rationale is to focus less on sleep-associated memory consolidation and instead focus on the literature which has found that sleep disruption prior to learning is detrimental for memory formation (for example, see this meta-analysis: https://doi.org/10.1037/bul0000348). The Authors have included a sentence on this, but the take-home message of that paragraph seems to be that sleep disruption after learning is detrimental for consolidation.

Response: We have edited the paragraph in the introduction to include more about sleep disruption prior to learning: 

Research has shown that healthy sleep supports memory functioning in two ways: a good night’s sleep helps prepare the brain for new learning the following day, and sleep-dependent memory consolidation helps strengthen new memories learned during the day (Newbury et al., 2021). Sleep plays an important role in restoring neuronal functioning for new learning on the subsequent day (Tononi & Cirelli, 2003). Several studies have confirmed this and shown that when sleep is deprived, memory encoding is disrupted (e.g., Alberca-Reina et al., 2015). Studies also suggest that an uninterrupted night of sleep prior to learning allows for more efficient retrieval of that information on subsequent occasions (Cordeira et al. 2018; Walker 2008).

Our sincere thanks indeed.

Dr Michelle Henry

---

## [Decision Letter · Decision Letter 2]

13 Dec 2023

Sleep and Cognition in South African Patients with Non-Functioning Pituitary Adenomas

PONE-D-23-02515R2

Dear Dr. Henry,

Thank you for submitting your revised manuscript addressing the reviewers' comments.

I am pleased to inform you that, after the second round of reviews, your manuscript has been judged scientifically suitable for publication and will be formally accepted for publication once it meets all outstanding technical requirements.

Kind regards,

Ioanna Markostamou, Ph.D.

Academic Editor

PLOS ONE

Reviewers' comments:

Reviewer's Responses to Questions

**Comments to the Author**

1. If the authors have adequately addressed your comments raised in a previous round of review and you feel that this manuscript is now acceptable for publication, you may indicate that here to bypass the “Comments to the Author” section, enter your conflict of interest statement in the “Confidential to Editor” section, and submit your "Accept" recommendation.

Reviewer #3: All comments have been addressed

2. Is the manuscript technically sound, and do the data support the conclusions?

Reviewer #3: (No Response)

3. Has the statistical analysis been performed appropriately and rigorously? 

Reviewer #3: (No Response)

4. Have the authors made all data underlying the findings in their manuscript fully available?

Reviewer #3: (No Response)

5. Is the manuscript presented in an intelligible fashion and written in standard English?

Reviewer #3: (No Response)

6. Review Comments to the Author

Reviewer #3: (No Response)

7. PLOS authors have the option to publish the peer review history of their article (what does this mean?). If published, this will include your full peer review and any attached files.

Reviewer #3: No

---

## [Editor Report · Acceptance letter]

4 Jan 2024

PONE-D-23-02515R2 

PLOS ONE

Dear Dr. Henry, 

I'm pleased to inform you that your manuscript has been deemed suitable for publication in PLOS ONE. Congratulations! Your manuscript is now being handed over to our production team.

Kind regards, 

on behalf of

Dr. Ioanna Markostamou 

Academic Editor

PLOS ONE